# Microencapsulation of Carvacrol by Complex Coacervation of Walnut Meal Protein Isolate and Gum Arabic: Preparation, Characterization and Bio-Functional Activity

**DOI:** 10.3390/foods11213382

**Published:** 2022-10-27

**Authors:** Jishuai Sun, Yishen Cheng, Tuo Zhang, Jiachen Zang

**Affiliations:** Beijing Key Laboratory of Functional Food from Plant Resources, College of Food Science and Nutritional Engineering, China Agricultural University, Beijing 100083, China

**Keywords:** carvacrol, complex coacervation, walnut meal protein isolate, gum Arabic, bio-functional activity

## Abstract

As a natural phenolic compound, carvacrol has attracted much attention due to its excellent antibacterial and antioxidant activities. However, its application is limited due to its instability, such as easy volatilization, easy oxidation, etc. Protein-polysaccharide interactions provide strategies for improving their stability issues. In this study, the plant-based carvacrol microcapsules via complex coacervation between walnut meal protein isolate (WMPI) and gum Arabic (GA) has been fabricated and characterized. The formation conditions of WMPI-GA coacervates were determined by some parameters, such as pH, zeta-potential, and turbidity. The optimum preparation conditions were achieved at pH 4.0 with a WMPI-to-GA ratio of 6:1 (*w*/*w*). The mean particle size, loading capacity (*LC*), and encapsulation efficiency (*EE*) of the microcapsules were 43.21 μm, 26.37%, and 89.87%, respectively. Fourier transform infrared spectroscopy (FT-IR) and fluorescence microscopy further confirmed the successful microencapsulation of carvacrol. The microencapsulation of carvacrol improved the thermal stability of the free carvacrol. The swelling capacity results indicated that it could resist gastric acid, and facilitate its intestinal absorption. Meanwhile, the carvacrol molecules trapped within the microcapsules could be continuously released in a concentration-dependent manner. Furthermore, the microcapsules presented good antioxidant activity and antibacterial activity against the Gram-negative (*E. coli*) and the Gram-positive (*S. aureus*) bacteria. These results indicated that the obtained carvacrol microcapsules have a potential application value as a food preservative in the food industry.

## 1. Introduction

Carvacrol (5-Isopropyl-2-methylphenol), a major component of oregano and thyme oils, has a wide-spectrum antibacterial activity and good antioxidant activity [1,2,3]. It has been approved as a safe food preservative and food flavoring supplement used in food industry by Food and Drug Administration (FDA), and Generally Recognized as Safe (GRAS) [4,5]. However, several negative factors severely limit the applicability of carvacrol in the food industry, such as high sensitivity, high volatility, poor water solubility and low stability [1]. In recent years, some protective strategies for carvacrol have been reported, such as active films [3,6], nanocomplexes [4,7], emulsion [8], and microencapsulation [1], etc.

In these above protection strategies, complex coacervation represents an uncomplicated and promising method of microencapsulation of bioactive compounds [9,10]. It is also a popular approach for its simplicity, low cost, scalability and reproducibility in encapsulation of active compounds [11]. It usually refers to the formation of a stable mixture between two oppositely charged biopolymers (such as polysaccharides and proteins) through electrostatic interactions under certain conditions [10,12]. According to previous studies, gum Arabic (GA) and gelatin were the most commonly used biopolymers for the preparation of microcapsules by complex coacervation [13,14,15,16]. GA is a negatively charged hetero-polysaccharide, which is negatively charged in the range of pH 3–6. However, the charge of gelatin gradually increased with the decrease of pH value, and crossed the zero point of potential at an approximate pH of 4.8, that is, the isoelectric point of gelatin. The basis of the complex coacervation is the electrostatic interaction between GA and gelatin. Moreover, GA has good film-forming and emulsifying properties, which can effectively encapsulate active ingredients, especially for some volatile active ingredients, which can effectively prevent their volatilization and reduce their oxidation [16,17]. Currently, the utilization of gelatin has some negative aspects, such as high cost, the spread of drug-resistant pathogens, animal diseases and adverse environmental impact [18,19]. Furthermore, some people do not eat meat for religious reasons or due to veganism [20]. Therefore, it is necessary to find safer and cheaper plant protein resources to replace gelatin.

Walnut meal protein isolate (WMPI) is a by-product of the walnut oil extraction process, which is usually used as animal feed or discarded, resulting in a serious waste of resources and adverse environmental impacts [21,22]. Notably, except for methionine, WMPI has a balanced content of essential amino acids, which is within the range of the Food and Agriculture Organization (FAO) recommended model for adults [21,23]. According to previous studies, some plant protein isolates (such as barley protein, pea protein isolate, soybean protein isolate, brown rice protein, hemp protein, and sunflower protein, etc.) were usually used as the shell materials for microencapsulation [24,25]. Additionally, previous studies have reported that the solubility and emulsifying properties of walnut protein isolate can be improved by uncomplicated physical or chemical modification [22,26]. Therefore, as a sustainable and high-quality plant protein, WMPI is a promising alternative to gelatin as a wall material for microencapsulation of carvacrol by complex coacervation and for use in the food industry. Chen et al. (2021) [19] have reported that eugenol encapsulated microcapsules were developed using quinoa protein (QP) and GA as wall materials via complex coacervation, but their bio-functional activity were not further studied. Similarly, Qiu et al. (2021) [18] only reported the release properties of microcapsules but did not evaluate their bio-functional activities. Therefore, it is necessary to develop a novel multifunctional microcapsule and evaluate their bio-functional activities. To the best of our knowledge, no relevant studies on preparation and evaluation of carvacrol microcapsules using WMPI and GA by complex coacervation have so far been reported.

The objectives of this study were to explore optimum formation conditions of WMPI-GA coacervates and to prepare carvacrol microcapsules using the optimum ratio of WMPI and GA through complex coacervation. The proposed scheme of the preparation of the carvacrol encapsulated microcapsules is shown in Figure 1. Firstly, the optimum formation conditions of WMPI-GA coacervates were determined. Then, the physicochemical properties and micromorphology of carvacrol microcapsules were characterized and evaluated. Finally, the sustained release performance, antioxidant capacity and antibacterial capacity of carvacrol microcapsules were evaluated.

## 2. Materials and Methods

### 2.1. Materials and Chemicals

Walnut meal (by-products of walnut oil extraction) was provided by a walnut oil Food Co., Ltd. (Xinjiang, China). Carvacrol was purchased from Aladdin Chemical Reagent Co. Ltd. (≥99% purity, Shanghai, China). Gum Arabic (GA), Nile Red, n-hexane and potassium bromide (KBr) were purchased from Macklin Biotechnology Co., Ltd. (Shanghai, China). 2,2-diphenyl-1-picrylhydrazyl (*DPPH*) was purchased from Sigma Chemical Reagent Co., Ltd. (Shanghai, China). *Escherichia coli ATCC 25922* (*E. coli*) and *Staphylococcus aureus ATCC 6538* (*S. aureus*) were provided by the food microbiology laboratory in the College of Food Science and Nutritional Engineering, China Agriculture University (Beijing, China). All of the other chemical reagents were of analytical grade and purchased from Sino-pharm Chemical Reagent Co., Ltd. (Shanghai, China).

### 2.2. Extraction of WMPI

WMPI was extracted according to Lei et al. (2012) [21] with slight modifications. Briefly, walnut meals were defatted by using n-hexane. The defatted walnut meal flour solutions (10%, *w*/*v*) were continuously stirred for 6 h at 45 °C to fully hydrate. Then, the pH value of the solutions was adjusted to 11 using 1.0 M NaOH. Resultant solutions were continuously stirred for 2 h, followed by centrifugation at 10,000 r/min for 10 min at 4 °C. Subsequently, the supernatant was adjusted to pH 4.5 using 1.0 M HCl, followed by centrifugation at 10,000 r/min for 10 min at 4 °C. The obtained precipitate was reconstituted in deionized water at pH 7.0 and then dialyzed for 72 h. The samples were obtained by using freeze-drying and stored at −20 °C until use.

### 2.3. Preparation of WMPI and GA Solutions

The freeze-dried WMPI powder was dissolved in deionized water and continuously stirred for 1 h at room temperature. Then, the solution was adjusted to pH 11 using 1.0 M NaOH solution and continuously stirred for 2 h at room temperature. Subsequently, the solution was adjusted to pH 7.0 using the acetic acid solution of different concentrations (0.1%, 1%, and 10%, *v*/*v*) and continuously stirred overnight at 4 °C. In addition, GA powder was dissolved in deionized water with stirring for 2 h at room temperature. Finally, the above-obtained solutions were stored at 4 °C until use.

### 2.4. Effect of pH on the Zeta Potential of WMPI and GA Solutions

The zeta potential of WMPI and GA solutions was determined in the range of pH 3.0 to 6.0 by using a Zetasizer nano ZSE (Zeta ZEN3700, Malvern Instruments, Malvern, UK).

### 2.5. Effect of the WMPI to GA Ratio on the Formation of the WMPI-GA Complex

To determine the optimum ratio of the WMPI to GA, a series of WMPI/GA solutions (0.1%, *w*/*v*) were mixed at different ratios (WMPI: GA = 1:1, 2:1, 4:1, 6:1, and 8:1, *w*/*w*), respectively. After mixing, the WMPI/GA mixture was adjusted to pH 4.0 using 1.0 M NaOH or acetic acid solution of different concentrations (0.1%, 1%, and 10%, *v*/*v*). The turbidity and the zeta potential of the mixture at different ratios (WMPI: GA = 1:1, 2:1, 4:1, 6:1, and 8:1, *w*/*w*) were determined by using a UV-vis spectrophotometer and a Zetasizer nano ZSE, respectively.

### 2.6. Preparation of the Carvacrol Encapsulated Microcapsules

The carvacrol encapsulated microcapsules were prepared according to Chen et al. (2021) [19] with slight modifications. Briefly, WMPI solution (1%, *w*/*v*) and GA solution (1%, *w*/*v*) were mixed at the mass ratio of 6:1 under stirring for 2 h at room temperature. An appropriate amount of carvacrol (carvacrol: biopolymers = 2:1, based on the total added weight of biopolymers, *w*/*w*) was added to the above mixed solution. Then, the above-mixed solution was homogenized at 10,000 r/min for 5 min using a high shear homogenizer (T25 digital ULTRA-TURRAX^®^, IKA, Staufen, Germany). The above obtained mixed solution was adjusted to Ph 4.0 using the acetic acid solution of different concentrations (0.1%, 1%, and 10%, *v*/*v*) under stirring at 200 r/min. Finally, the solution was stirred at 100 r/min for 0.5 h in an ice bath to facilitate the formation of the carvacrol encapsulated microcapsules. The obtained carvacrol encapsulated microcapsules were left to stand overnight at 4 °C.

The control group produced without adding carvacrol was named “WMPI-GA coacervates”. WMPI-GA coacervates were prepared in the same way as the carvacrol encapsulated microcapsules preparation above. The above obtained WMPI-GA coacervates and carvacrol encapsulated microcapsules were freeze-dried and stored at −20 °C until analysis.

### 2.7. Characterization of WMPI-GA Coacervates and Carvacrol Microcapsules

#### 2.7.1. Mean Particle Size

For the liquid samples before freeze-drying, the particle size and particle size distribution were determined by using a laser scattering size analyzer (LS230^®^, Beckman Coulter, Beijing, China). Firstly, the refraction indices of the samples and water were 1.59 and 1.33, respectively. Then, the liquid samples were added into the laser scattering size analyzer. Finally, the particle size and size distribution were recorded by the software. All measurements were tested at least three times in parallel.

#### 2.7.2. Morphology

For the liquid samples before freeze-drying, the morphology of WMPI-GA coacervates and carvacrol encapsulated microcapsules was captured using upright fluorescence microscopy (Leica DM6 B, Leica Microsystems, Shanghai, China) at a magnification of 200×. To further observe the presence of carvacrol in the microcapsules, the carvacrol encapsulated microcapsules were stained by Nile Red (0.2%, *w*/*v*) solution for 15 min. Then, the stained samples were dropped onto a slide. Finally, the stained samples were observed by upright fluorescence microscopy at a magnification of 200×.

#### 2.7.3. Encapsulation of Carvacrol in the Microcapsules

Encapsulation Efficiency (*EE*) and loading capacity (*LC*) were evaluated as recently reported by Chen et al. (2021) [19]. Briefly, the freeze-dried carvacrol microcapsules (20.0 mg) were soaked in 20 mL of ethanol. Then, they were gently shaken at 150 rpm for 5 min and the supernatant was collected by centrifugation at 10,000 r/min for 10 min for the determination of surface carvacrol (*SC*). The obtained supernatant was used to determine the concentration of *SC* at 275 nm with the UV-vis spectrophotometer. The total carvacrol (*TC*) of the microcapsules was obtained by using the sonication for 0.5 h at 40 °C. After centrifugation at 10,000 r/min for 15 min at 25 °C, the pellet was reconstituted and then sonicated in the same method as above. The above treatment was carried out at least three times until rinsed. All obtained above solution was used to determine the total concentration of TC at 275 nm with the UV-vis spectrophotometer. The content of carvacrol was calculated based on the calibration curve (y = 17.031x − 0.0926, R^2^ = 0.998; where x was the concentration of carvacrol, y was the absorbance at 275 nm). Encapsulation Efficiency (*EE*) and loading capacity (*LC*) were calculated by the following formulas:(1)$$\mathrm{EE}\;(\%)=\frac{W_{\mathrm{TC}}{\;-\;W}_{\mathrm{SC}}}{W_{\mathrm{TC}}}\;\times \;100$$
(2)$$\mathrm{LC}\;(\%)=\frac{W_{\mathrm{TC}}}{W_{\mathrm{MS}}}\;\times \;100$$
where *W_MS_* was the weight of the microcapsule sample. *W_TC_* and *W_SC_* were the weight of *TC* and *SC* of carvacrol in the microcapsule, respectively.

#### 2.7.4. Swelling Capacity

The swelling capacity was evaluated according to Wu et al. (2021) [27] with slight modifications. For gastric fluids, dilute HCl with a concentration of 1 mol/mL was diluted with water, and the pH was adjusted to 2.0 and 4.0. For intestinal fluids, 6.8 g of KH_2_PO_4_ was dissolved in deionized water (500 mL). Then, the pH of the solution was adjusted to 7.5 using 0.1 M NaOH aqueous solution. Specifically, the freeze-dried samples (50.0 mg) were soaked in simulated gastrointestinal fluids (pH 2.0, pH 4.0 and pH 7.5) at 25 °C for 12 h, respectively. Then, the mixtures were centrifuged at 10,000 r/min for 10 min to evaluate their swelling capacity. Swelling capacity can be calculated by the following equations:(3)$$\mathrm{Swelling}\;\mathrm{capacity}=\frac{W_{p}}{W_{0}}$$
where *W_p_* was the weight of the precipitate after centrifugation of the sample and *W*_0_ was the initial weight of the freeze-dried samples.

#### 2.7.5. FT-IR Spectroscopy

The molecular structure and chemical bonds of carvacrol, GA and other freeze-dried samples (WMPI, WMPI-GA coacervates and carvacrol encapsulated microcapsules) were determined with FT-IR spectrometer (Thermo Fisher Scientific Co., Ltd., Waltham, MA, USA). The samples were mixed with KBr at a ratio of 1:100 and ground quickly and evenly with an agate mortar. The FT-IR worked at a resolution of 4 cm^−1^, and 32 scans were done in the range of 4000–400 cm^−1^.

#### 2.7.6. Thermal Gravimetric Analysis (TGA)

The thermal stability of free carvacrol, freeze-dried WMP-GA coacervates, and freeze-dried carvacrol microcapsules was evaluated by using a thermo-gravimetric analyzer (STA409-PC, Netzsch, Shanghai, China). The solid sample weighs approximately 5.0 mg. The analytical instrument was at a heating rate of 20 °C/min from 40 °C to 500 °C under a nitrogen atmosphere at a scan rate of 20 mL/min.

#### 2.7.7. Carvacrol Release Property

In the release experiment, 50% ethanol solution was used as the food simulant, which is often used as the simulated fat food [19]. Briefly, a certain amount of the freeze-dried carvacrol microcapsules (50.0 mg) were dissolved in 50 mL of the food simulants and stirred (150 rpm) for 24 h at 25 °C under dark condition. At every time interval, taking a certain volume of the sustained-release solution and adding an equal volume of food simulant. The amount of carvacrol released was calculated according to the standard curve in Section Section 2.7.3. The cumulative release (%) was calculated by the following formulas:(4)$$\mathrm{Cumulative}\;\mathrm{release}\;\left(\%\right)=\frac{M_{t}}{M_{0}}\;\times \;100$$
where *M_t_* and *M*_0_ were the mass of carvacrol released from the carvacrol microcapsule at time and the total mass of carvacrol encapsulated in the carvacrol microcapsule, respectively.

To further clarify the release mechanism of carvacrol in the food simulation, the above cumulative release curve was fitted by using the Zero-order model, First-order model, and Higuchi model, respectively.

#### 2.7.8. Antioxidant Capacity

The antioxidant capacity of the samples was evaluated by *DPPH* free radical scavenging assay. Briefly, an equal amount of carvacrol, WMP-GA coacervates and carvacrol microcapsules were individually dissolved in 10 mL of ethanol. The above solutions were placed for 12 h in the dark at 4 °C. Then, 2 mL of sample solution was added to 2 mL of 100 μM *DPPH* solution and reacted for 0.5 h in the dark. An equal amount of ethanol was regarded as the control. Finally, the absorbance of the samples was measured at 517 nm and calculated as follows:(5)$$\mathrm{DPPH}\;\mathrm{scavenging}\;\mathrm{ability}\;(\%)=\frac{A_{c}{\;-\;A}_{s}}{A_{c}}\;\times \;100$$
where *A_c_* was the absorbance of the control and *A_s_* was the absorbance of the samples.

#### 2.7.9. Antibacterial Capacity

The antibacterial capacity of the samples against *S. aureus* and *E. coli* bacteria was evaluated according to Sun et al. (2020) [28] with slight modifications. Firstly, 100 μL of prepared bacterial suspension of the inoculums (10^6^~10^7^ CFU/mL) was spread evenly on LB agar. Then, the freeze-dried samples (5.0 mg) were compressed in powder form into 12.0 mm diameter and placed on the above inoculated LB agar. Finally, the Petri dishes were placed upside down and incubated at 37 °C for 12 h and 24 h; the diameter of the inhibition zone was measured after 24 h.

In addition to using the agar disk diffusion method with the judgement of the inhibition zones (mm), the following evaluation was conducted to further clarify the potential antimicrobial activity of the samples against *S. aureus* and *E. coli* bacteria. Briefly, 50 mg of the freeze-dried samples and 100 μL bacterial suspension (10^5^ CFU/mL) were added into a 10 mL nutrient broth and then cultured in a constant temperature shaker under 37 °C at 100 rpm for 2 h. Then, 100 μL of the above solution was spread evenly on LB agar. Finally, the Petri dishes were placed upside down, incubated at 37 °C for 12 h and 24 h, and the growth of bacterial colonies was observed.

### 2.8. Statistical Analyses

Statistical data were analyzed using Origin^®^ Pro 9.1 software (Origin Lab, Northampton, MA, USA) and SPSS^®^ 25.0 software (SPSS Inc., Chicago, IL, USA). Least significant differences (LSD) multiple comparison tests were used to determine the significance of the obtained data (*p* ˂ 0.05). All tests were conducted in triplicate and all obtained data were presented as mean ± standard deviation.

## 3. Results

### 3.1. Construction of the Complex Coacervation System between WMPI and GA

#### 3.1.1. Effect of pH on the Formation of WMPI and GA Complex

Complex coacervation is usually driven by electrostatic interactions between oppositely charged biopolymers, such as the typical combination of protein and gum [16,19,29]. In the process of complex coacervation, exploring the effect of pH on the zeta potential of biopolymers is critical for optimizing the preparation of complex coacervate systems [18,19]. Specifically, the higher the absolute value of the potential difference between the oppositely charged biopolymers, the stronger the electrostatic interaction between two biopolymers, and the more conducive to the formation of a stable complex coacervate system [29]. To prepare a stable complex coacervate system, the zeta potential (surface charge) of WMPI and GA was measured in the range of pH 3.0–5.5, respectively. As shown in Figure 2, the zeta potential of WMPI ranged from 21.53 mV (pH 3.0) to −10.43 mV (pH 5.5) due to the existence of amino (-NH2) and carboxyl (-COOH) groups on the WMPI structure. Notably, the zeta potential of WMPI was close to 0 mV at an approximate pH of 4.7, which was defined as the isoelectric point (pI, electrically neutral point) of WMPI [19]. In contrast, the zeta potential of GA was all negative from 23.77 mV (pH 5.5) to 7.93 mV (pH 3.0) in the range of pH 3.0–5.5. This result is related to the fact that GA is a negatively charged hetero-polysaccharide [16]. Therefore, the complex coacervation between WMPI and GA can only occur below the pI of WMPI. It was found that WMPI and GA had a higher absolute value (34.8 mV) of potential difference at pH 4.0, which is beneficial to the formation of a stable complex system.

#### 3.1.2. Effect of a WMPI to GA Ratio at pH 4.0 on the Formation of the Complex

Besides pH, a mixing ratio of WMPI and GA is also a key parameter affecting the formation of stable complex coacervate systems [19]. The zeta potential of WMPI/GA mixtures at different mixing ratios (1:1, 2:1, 4:1, 6:1, and 8:1, *w*/*w*) is displayed in Figure 3A. It was observed that the zeta potential of the WMPI/GA mixture with a ratio at 6:1 (*w*/*w*) was lowest and tended to be electrically neutral compared to that of other ratios, suggesting that the mixing ratio of 6:1 between WMPI and GA was an optimum ratio for the formation of stable complex coacervate systems. According to Figure 3B, the turbidity of WMPI-GA systems with different mixing ratio was determined in the range of pH 3.0–5.5. Considering pI of WMPI, the optimum pH for complex coacervation of WMPI-GA systems with various ratios (1:1, 2:1, 4:1, 6:1, and 8:1, *w*/*w*) was 4.0. To further clarify the optimal ratio, the turbidity of WMPI/GA mixtures with different mixing ratios (1:1, 2:1, 4:1, 6:1, and 8:1, *w*/*w*) at pH 4.0 was determined and results are presented in Figure 3C. It was observed that the turbidity of the WMPI/GA mixture at a ratio of 6:1 was higher than that of the mixture at a ratio of 4:1, 2:1, and 1:1, and this might be derived from the overloading of GA, which causes steric repulsion, thereby reducing turbidity. Interestingly, the turbidity of the WMPI/GA mixture at an 8:1 ratio was lower than the mixing ratio of 6:1, which was probably related to the overloading of WMPI that could not significantly increase the turbidity of the mixture. Similar results were reported in the previous studies [19,30]. Given the above results, it can be concluded that the optimum preparation conditions for complex coacervation between WMPI and GA were pH 4.0 and the WMPI-GA mixing ratio of 6:1 (*w*/*w*).

### 3.2. Size Distribution and Morphology of WMPI-GA Coacervates and Carvacrol Microcapsules

Having determined the optimum preparation conditions for complex coacervation between WMPI and GA, we encapsulated carvacrol molecules within microcapsules formed by the above WMPI-GA coacervates based on such conditions. To confirm such encapsulation, the inverted fluorescence microscope and laser scattering size analyzer was respectively used to observe the morphology and size distribution of the resulting liquid coacervates and carvacrol microcapsules. The size distribution and several representative morphology images are shown in Figure 4. The liquid WMPI-GA coacervates had rough and irregular network shape with a mean particle size of 47.42 μm (Figure 4A,C). The liquid carvacrol microcapsules had dense and irregular network shape with a mean particle size of 43.21 μm (Figure 4B,C). It can be seen that the carvacrol microcapsules exhibited a layer-by-layer core structure (Figure 4B), suggesting that carvacrol was successfully encapsulated within the prepared microcapsules. Agreeing with this view, similar layer-by-layer core structures have already been reported with the microcapsules [19,31]. Moreover, it can be found that the mean particle size decreased from 47.42 μm to 43.21 μm after microencapsulation (Figure 4C), which was exactly consistent with the dense network shape of the carvacrol microcapsules. To further confirm the state of carvacrol in the microcapsule structure, the morphology of the carvacrol microcapsules stained by Nile Red is presented in Figure 4D under an upright fluorescence microscope. It was found that the carvacrol in the structure of the carvacrol microcapsules was stained red by Nile Red. Almost no free stained carvacrol was distributed in the dark background, which also indicated that the carvacrol was successfully encapsulated in the microcapsules.

### 3.3. Encapsulation Efficiency and Swelling Capacity of the Carvacrol Microcapsules

The encapsulation efficiency (*EE*) and loading capacity (*LC*) of the carvacrol microcapsules prepared by using WMPI-GA complex coacervate as shell materials were also evaluated and the results are shown in Figure 5A. The *EE* determines the percentage of carvacrol that is encapsulated in the WMPI-GA coacervates matrix compared with total carvacrol including SC and encapsulated carvacrol, and a higher *EE* is better. The *LC* measures the percentage of carvacrol content in total freeze-dried microcapsules powder. The *EE* and *LC* of the carvacrol microcapsules were 89.87 ± 0.42% and 26.37 ± 1.28%, respectively, indicating that the wall materials of using WMPI-GA complex coacervates can effectively microencapsulate carvacrol under the present preparation conditions. Similarly, Chen et al. (2021) [19] obtained a higher *EE* (88.60%) at a ratio of 4:1 in quinoa protein/GA microcapsules containing eugenol. However, the relatively low *LC* (26.37%) depends on the concentration of carvacrol added, which is consistent with the results of this study. Therefore, we can find that there is no direct relationship between *EE* and *LC*. Moreover, the swelling capacity of the WMPI-GA coacervates and carvacrol microcapsules in simulated gastrointestinal fluids (pH 2.0, pH 4.0, and pH 7.5) in vitro are shown in Figure 5B. It can be found that the swelling capacity of WMPI-GA coacervates and carvacrol microcapsules was stable at pH 4.0. The swelling capacity of carvacrol microcapsules was lowest at pH 2.0 compared to WMPI-GA coacervates, which may be attributed to the dense structure of the carvacrol microcapsules [27]. GA, a negatively charged hetero-polysaccharide, is negatively charged in the range of pH 2–6. However, the charge of WMPI gradually increased with the decrease of pH value, and crossed the zero point of potential at approximately pH 4.7. It can be seen that WMPI and GA are more conducive to forming a dense structure at pH 2.0. Notably, the swelling capacity of carvacrol microcapsules significantly increased at pH 7.5 compared to other pH conditions, indicating that carvacrol was more easily released from carvacrol microcapsules under pH 7.5. The above results indicated that carvacrol encapsulated microcapsules could resist gastric acid, thereby facilitating its intestinal absorption [27].

### 3.4. FT-IR Analysis

The FT-IR spectra of WMPI, GA, carvacrol, WMPI-GA coacervates, and carvacrol microcapsules are shown in Figure 6. For the FTIR spectrum of WMPI, a broad peak was observed at 3294 cm^−1^ due to the O–H group stretching of the free amino acid [18]. Three peaks at 1657 cm^−1^ (Amide I), 1537 cm^−1^ (Amide II) and 1236 cm^−1^ (Amide III) mainly corresponded to C=O stretching, N–H bending, and C–N stretching, respectively. The above typical peaks indicated the presence of a large fraction of protein [32]. GA is a heteropolysaccharide that consists of galactose and glycoproteins [33]. The FTIR spectrum of GA showed a broad peak at 3389 cm^−1^, which was attributed to the amino groups (Amide A). The peaks at 1606 cm^−1^ and 1420 cm^−1^ corresponded to the asymmetrical and symmetrical stretching vibrations of –COOH groups with the negative charge [33].

Compared to the individual FT-IR spectrum of WMPI and GA, the FTIR spectrum of WMPI-GA coacervates appeared to be a combined spectrum consisting of WMPI and GA, but it was more similar to the FTIR spectrum of WMPI. In the FTIR spectrum of WMPI-GA coacervates, the peak of Amide I of WMPI shifted from 1657 cm^−1^ to 1659 cm^−1^, while the peaks of typical –COOH groups (1606 cm^−1^ and 1420 cm^−1^) of GA disappeared, indicative of the presence of electrostatic interactions between WMPI and GA [30,34].

The FT-IR spectrum of free carvacrol exhibited two absorption peaks at 3347 cm^−1^ and 2960 cm^−1^ attributed to O–H and C=C–C groups, respectively. The peaks at 2926 cm^−1^ and 2869 cm^−1^ corresponded to the asymmetric and symmetric C–H stretching of methylene [18]. The peaks at 1620 cm^−1^ and 1420 cm^−1^ corresponded to C–C groups of aromatic rings of carvacrol [35]. The peaks at 864 cm^−1^ and 810 cm^−1^ were attributed to the typical aromatic ring of carvacrol [35]. As compared to the FTIR spectrum of WMPI-GA coacervates, the spectrum of carvacrol microcapsules at 2961 cm^−1^, 869 cm^−1^, and 810 cm^−1^ displayed novel absorption bands, which indicated that the microencapsulation of carvacrol was achieved. Furthermore, the O–H stretching and shrinking/widening of absorption bands at 3200 cm^−1^–3500 cm^−1^ indicates the formation of weak hydrogen bonds between carvacrol and coacervates [27,30].

### 3.5. Thermal Stability

Thermal gravimetric analysis (TGA) is used to evaluate the thermal stability of the samples. The differential thermogravimetric (DTG) curve is the first derivative of the TGA curve. The TGA and DTG curves of free carvacrol, WMPI-GA coacervates, and carvacrol microcapsules are presented in Figure 7A and Figure 7B, respectively. The TGA curve of the free carvacrol had a weight loss of 3.75% at the first stage (40–100 °C), which was attributed to the water evaporation and the volatilization of a small amount of carvacrol [19,35]. At the second stage (100–175 °C), the TGA and DTG curves of carvacrol presented a fast weight loss (approximately up to 94.48%), which indicated that carvacrol had been completely volatilized in the stage. For WMPI-GA coacervates, a slow weight loss of 12.40% at the first stage (40–250 °C), was attributed to the water evaporation and the decomposition of WMPI and GA. In the second stage, a rapid weight was lost from 250 °C to 500 °C because of the considerable thermal degradation of the coacervates. Compared to the TGA curve of the free carvacrol, the rapid weight loss of carvacrol microcapsules did not begin from 100 °C due to the microencapsulation of WMPI and GA. The thermal stability of carvacrol can be improved by using complex coacervation. Our findings are consistent with previous studies on microcapsules prepared from gelatin and other biopolymers [14,36], which also suggest that WMPI is a promising alternative to gelatin as a wall material for microencapsulation of bioactive substances by complex coacervation.

### 3.6. Carvacrol Release from the Prepared Microcapsules

The release behavior of carvacrol from the carvacrol microcapsules in the food simulant was determined, and results are shown in Figure 8. The carvacrol release presented a burst release from 0 h to 2 h, followed by a relatively slow release from 2 h to 8 h. For carvacrol microcapsules, the burst release presented within the first 2 h was due to the surface and voids of the carvacrol microcapsules particles, while the carvacrol encapsulated in the interior presented a relatively slow release over time. Another possible reason is that carvacrol, which remains on the surface and in the voids of the microcapsules, is more soluble in oily food simulant. A similar release trend was also observed in the eugenol microcapsules [19]. At the same time, some studies have revealed that the encapsulated essential oil presented higher release properties in oily food products than other simulated foods [18,19]. The carvacrol release reached a plateau and kept sustained release after 8 h, suggesting that WMPI-GA as the microcapsule wall material can achieve sustained release of carvacrol. Additionally, the fitted release curves of carvacrol are shown in Figure 8B–D by using three kinetic equations (zero-order, first-order, and Higuchi). According to the R^2^ values of three kinetic equations, the R^2^ values were 0.1625 (zero-order), 0.9593 (first-order), and 0.4450 (Higuchi), respectively. Based on the obtained results, the first-order was more suitable for evaluating the carvacrol release [19].

### 3.7. Antioxidant Capacity of the Carvacrol Microcapsules

Carvacrol has relatively strong free radical scavenging activity due to its structure with phenolic isomers [37]. In this study, the most commonly used DPPH free radical scavenging test method was used to evaluate the antioxidant capacity of carvacrol microcapsules. As shown in Table 1, the WMPI-GA coacervates possessed the lowest *DPPH* scavenging ability (*p <* 0.05). The weak antioxidant capacity of the WMPI-GA coacervates may be due to the limited free radical scavenging ability of WMPI and GA. In contrast to the antioxidant capacity of free carvacrol, the antioxidant capacity of the carvacrol within microcapsules was slightly reduced but still showed strong free radical scavenging capacity. The above results indicated that the antioxidant activity of carvacrol was not significantly affected by the complex coacervation microencapsulation process. A similar observation was also obtained with previously reported microencapsulation of carvacrol in a pectin-alginate matrix [1].

### 3.8. Antibacterial Capacity of the Carvacrol in the WMPI-GA Matrix Microcapsules

Food microbial contamination is related to life safety, so it is very important to resist microbial infection [28]. Typical Gram-positive bacteria (*S. aureus, ATCC 6538*) and Gram-negative bacteria (*E. coli, ATCC 25922*) are usually selected to evaluate the antibacterial capacity of microcapsules by disk diffusion [28]. As presented in Figure 9A,B, the WMPI-GA coacervates hardly had antibacterial activity against *E. coli* and *S. aureus*, while the carvacrol microcapsules showed better antibacterial activity against *E. coli* and *S. aureus* as well as the free carvacrol. After 12 h of incubation at 37 °C, it can be found that a large and distinct zone of inhibition appeared against *E. coli* and *S. aureus* compared to WMPI-GA coacervates. Moreover, the carvacrol microcapsules still had relatively strong antibacterial activity against *E. coli* and *S. aureus* after 24 h of incubation at 37 °C. To further quantitatively evaluate the antibacterial capacity of carvacrol microcapsules, the diameter of inhibition zones against *E. coli* and *S. aureus* after 24 h of incubation at 37 °C were 27.86 ± 2.30 mm and 33.78 ± 1.70 mm, respectively. It was found that the carvacrol microcapsules were more effective against *S. aureus* than *E. coli*. Previous research has also revealed that carvacrol had excellent antibacterial capacity [1]. However, the diameter of inhibition zones of free carvacrol against *E. coli* and *S. aureus* after 24 h of incubation at 37 °C were 26.39 ± 1.98 mm and 31.96 ± 1.01 mm, respectively. One possible reason is that free carvacrol is volatile and especially unstable in a 37 °C thermostat, thus reducing its antimicrobial activity. The carvacrol microcapsule can avoid the rapid volatilization of carvacrol, and present a slow release effect on carvacrol, which can effectively inhibit the growth of bacteria. In addition, another antimicrobial assay also showed that the carvacrol microcapsules showed excellent antibacterial capacity. As shown in Figure 9C, it can be observed that the petri dishes of the control group were full of bacteria after 12 h, while the group mixed with carvacrol microcapsules had no bacterial growth. After 24 h of incubation at 37 °C, it was found that the petri dishes of the control group presented that a trend of denser colony growth (Figure 9D). However, a few colonies appeared in the group mixed with carvacrol microcapsules. According to the above results, the microencapsulation of carvacrol did not significantly affect the antibacterial capacity of the microcapsules, which showed great potential to be used as a food preservative in food industry.

## 4. Conclusions

This study mainly developed a novel shell material by complex coacervation between WMPI and GA and applied it to preparing the multifunctional carvacrol microcapsules. The carvacrol microcapsules prepared under the optimum conditions (pH 4.0, WMPI-to-GA ratio = 6:1) had dense morphology, favorable size (43.21 ± 1.38 μm), high encapsulation efficiency (89.87 ± 0.42%) and good thermal stability. The FTIR results further demonstrated the successful encapsulation of carvacrol in microcapsules and good compatibility between WMPI and GA. The carvacrol microcapsules had a sustained-release behavior in food simulants and a high swelling capacity at pH 7.5. Moreover, the carvacrol microcapsules also had strong antioxidant capacity and antibacterial capacity against the Gram-negative (*E. coli*) and Gram-positive (*S. aureus*) bacteria. These results indicated that the microencapsulation of carvacrol by complex coacervation of WMPI and GA endowed the microcapsules with sustainable antioxidant and antibacterial activity, which are expected to be used in the development of plant-based food preservatives in the future.

## Figures and Tables

**Figure 1 foods-11-03382-f001:**
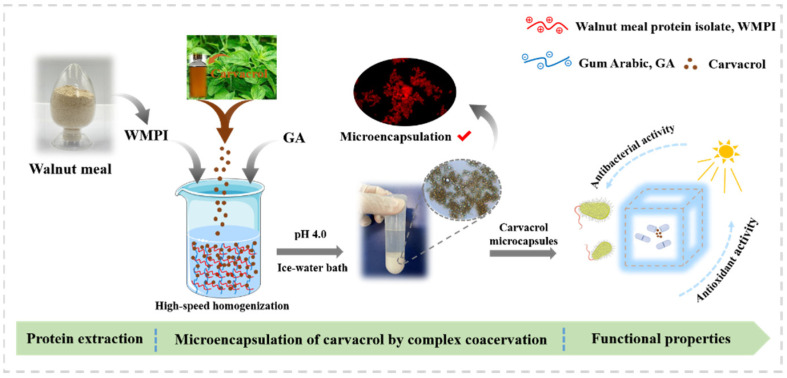
The proposed scheme for the preparation of the carvacrol encapsulated microcapsules.

**Figure 2 foods-11-03382-f002:**
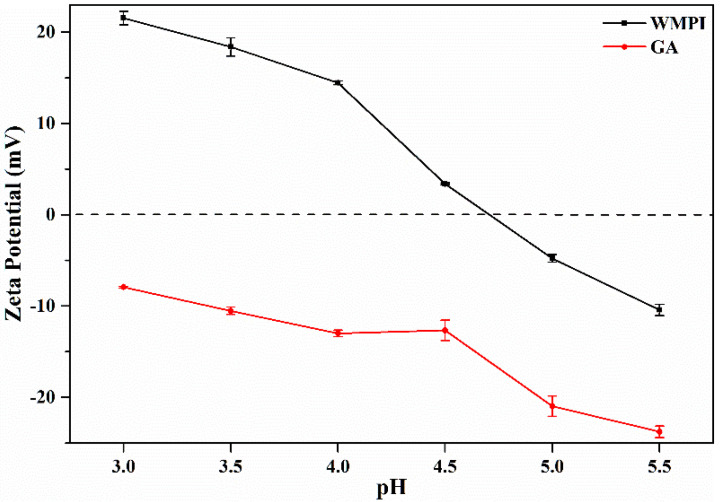
Effect of pH on zeta potential of WMPI and GA.

**Figure 3 foods-11-03382-f003:**
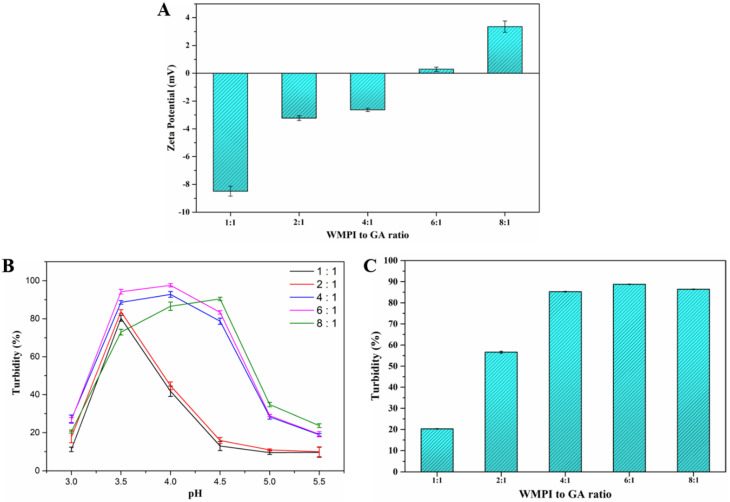
Effect of complex coacervation ratio of WMPI and GA at pH 4.0 on zeta potential (**A**), Turbidity of WMPI-GA systems with different mixing ratio as a function of pH (**B**) and turbidity of WMPI-GA systems with different mixing ratio at pH 4.0 (**C**).

**Figure 4 foods-11-03382-f004:**
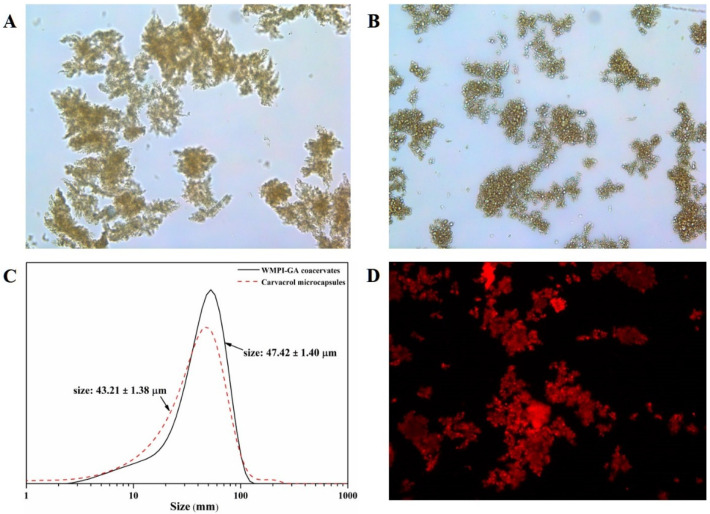
Upright fluorescence microscopic images of WMPI-GA coacervates and carvacrol microcapsules at the magnification of ×200. WMPI-GA coacervates and carvacrol microcapsules under bright field (**A**,**B**); Size distribution (**C**); The stained carvacrol microcapsules under fluorescence field (**D**).

**Figure 5 foods-11-03382-f005:**
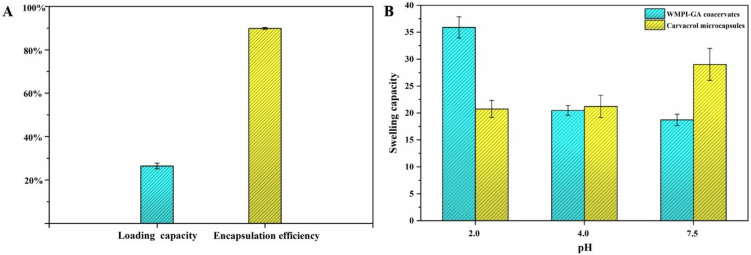
Encapsulation efficiency (**A**) and swelling capacity (**B**) of the carvacrol microcapsules.

**Figure 6 foods-11-03382-f006:**
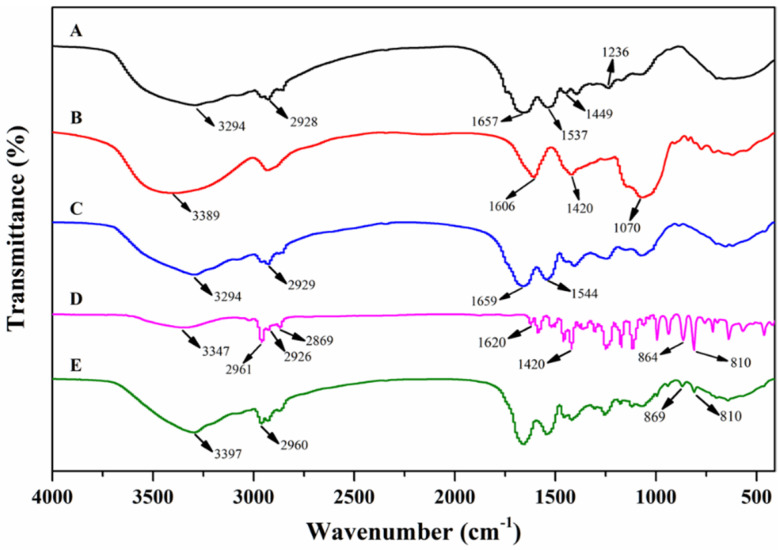
FTIR spectra of WMPI (**A**), GA (**B**), WMPI-GA coacervates (**C**), carvacrol (**D**), and carvacrol microcapsules (**E**).

**Figure 7 foods-11-03382-f007:**
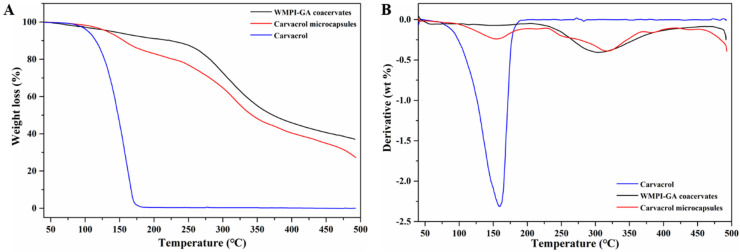
TGA (**A**) and DTG (**B**) curves of carvacrol, WMPI-GA coacervates, and carvacrol microcapsules.

**Figure 8 foods-11-03382-f008:**
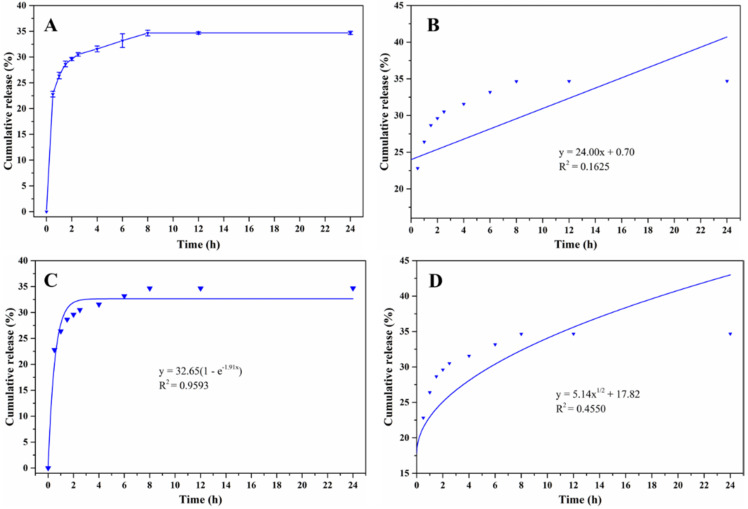
The cumulative release curve of carvacrol microcapsules in food simulant (**A**) and three release fitted curves: Zero-order (**B**), First-order (**C**), and Higuchi (**D**).

**Figure 9 foods-11-03382-f009:**
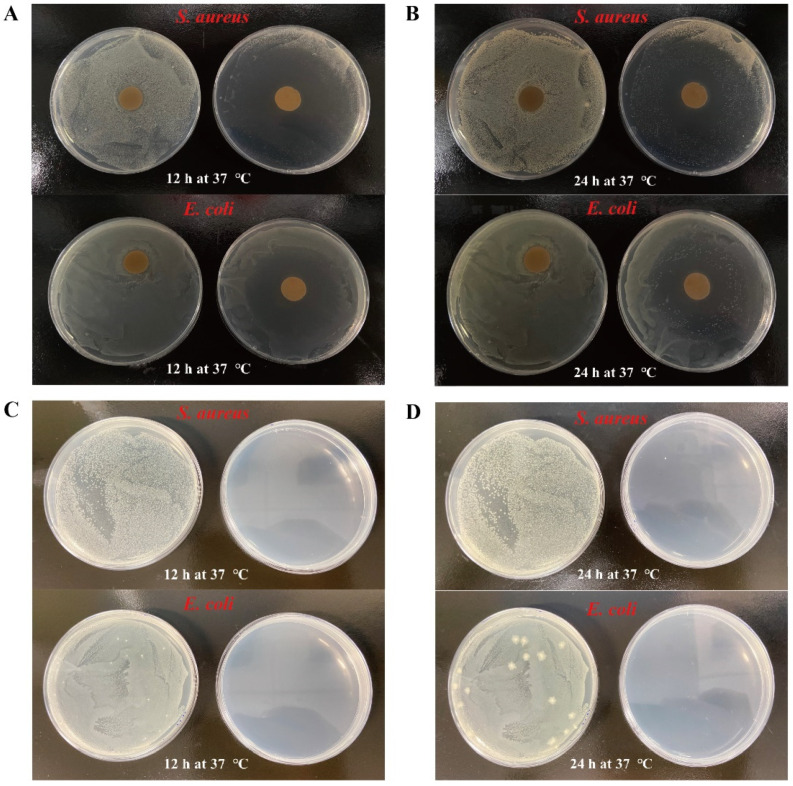
Digital photos of the inhibitory effect (**A**,**B**) and bacterial colonies (**C**,**D**) of *S. aureus* and *E. coli* after incubation with WMPI-GA coacervates and carvacrol microcapsules for 12 and 24 h.

**Table 1 foods-11-03382-t001:** Antioxidant and antibacterial capacity of the free carvacrol, WMPI-GA coacervates and carvacrol microcapsules.

Sample	*DPPH* Radical Scavenging Activity (%)	Diameter of Inhibition Zone (mm)	
		*S. aureus* (+)	*E. coli* (−)
Free carvacrol	54.47 ± 0.66	31.96 ± 1.01	26.39 ± 1.98
WMPI-GA coacervates	1.65 ± 0.55	---	---
Carvacrol microcapsules	48.51 ± 1.61	33.78 ± 1.70	27.86 ± 2.30

--- represents no effective antibacterial properties. All data are shown as mean ± standard deviation (SD).

## Data Availability

Data is contained within the article.

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
