# Peer review of "Microencapsulation of Carvacrol by Complex Coacervation of Walnut Meal Protein Isolate and Gum Arabic: Preparation, Characterization and Bio-Functional Activity"

_foods, 2022, doi:10.3390/foods11213382_

Round 1

Reviewer 1 Report

The submitted paper's content is compatible with the relevant special issue. The aim of the study is presented clearly. The experimental setup, procedure, and design are generally appropriate to achieve the goal. Although the work is quite simple, it is result-oriented and the study's experimental results can potentially contribute to the literature. There are some queries and problems that should be explained, discussed, and/or corrected in the manuscript (especially in the Materials & Methods section) to enhance the quality of the study:

1.     Lines 46-49: “GA, a negatively charged hetero-polysaccharide, is usually used as a wall material for volatile components…, excellent volatile retention, and inhibition of oxidation reactions”. This is a somewhat ambitious comment. GA is indeed a widely used wall material because of its properties (low cost, high solubility, unique emulsifying properties, etc). However, for such an interpretation of GA or any other wall material (to be suitable as a wall material in the encapsulation of a specific group of components and/or high volatile retention properties, inhibition of oxidation reactions, etc), the reasons should be explained with the mechanisms.

2.     The names of microorganisms should be written in italic form. This problem is encountered many times in the text. The whole manuscript should be checked and corrected.

3.     In Section 2.6, the production of coacervate, which is produced as a control group and does not have any encapsulation purpose, is explained, and then the production of carvacrol microcapsules is explained. This looks a little confusing. First of all, it would be more appropriate to explain the production of carvacrol microcapsules, which is the main purpose and sample in the study, and then to identify that the control group produced without adding carvacrol was called "WMPI-GA coacervates".

4.     What was used as a hardening agent in the production of coacervates? If not used, please explain the reason.

5.     The produced coacervates and microcapsules were finally freeze-dried. In some parts of the study, it was stated that the analyses were made in liquid forms. Were these liquid samples analyzed before freeze-drying or after reconstitution of powders? It should be clarified and, if reconstitution was made, the conditions/concentrations, etc for the reconstitution process should be written.

6.     Moreover, in some analyzes it is understood that the samples were examined in powder form. In the analysis, if the samples were in powder form, it is important to specify this for the clarity of the study. In addition, if we are talking about the analysis of powdered samples, it should be stated whether any grinding process is carried out after freeze-drying.

7.     The methods mentioned in sections 2.7.1 and 2.7.2 are not adequately described. It is necessary to give the details of the methods applied and to explain the conditions in which they were carried out.

8.     In section 2.7.3, the method was poorly described.

a.      Lines 148-149: “Briefly, the freeze-dried carvacrol microcapsules were soaked in 20 mL of ethanol”. How much powder was used?

b.     Lines 149-150: “The surface carvacrol (SC) of the microcapsules was obtained by shaking gently several times”. The sentence was not clear, it should be rewritten. Moreover, did the authors perform filtration or centrifugation? If so, what were the conditions?

c.      Lines 150-151: “The obtained solution was used to determine the concentration of SC at 275 nm with the UV-vis spectrophotometer.”. Please, explain this method and give references.

9.     In section 2.7.4, the methodology should be explained in detail. What is the composition of the GI? How much volume of GI was used? How much powder was used? How long and at what temperature was it treated? What were the centrifugation conditions?

10.  Please, give the details of the methodology explained in section 2.7.5.

11.  In section 2.7.6:

a.      Line 179: “food simulant was chosen to imitate fatty food models”. Is there a standard for this? Please, explain and give a reference.

b.     Lines 179-180: “Briefly, a certain amount of the freeze-dried carvacrol microcapsules were dissolved in 50 mL of food simulants…”. What was the “certain” amount? Please, be specific.

c.      How did the authors measure the variation of carvacrol?

12.  In section 2.7.9:

a.      Please, give the details (specific strains, etc) of the microorganisms.

b.     What was the concentration of the samples used in the analysis? Were they in liquid or powder form? Please, give details.

13.  Lines 227-228: “…10.43 mV (pH 6.0)…”. The value should be negative and the pH value should be 5.5 according to Figure 2.

14.  Lines 234-235: “It was found that WMPI and GA had a higher absolute value of potential difference at pH 4.0…”. Please, give the measured values at pH 4.0.

15.  Lines 297-299: Please, discuss the reason why the WMPI-GA coacervates swelled at pH 2.0.

16.  Lines 301-303: Please, give a reference for this comment/argument.

Reviewer 2 Report

The manuscript entitled « Microencapsulation of carvacrol by complex coacervation of walnut meal protein isolate and gum Arabic: Preparation, characterization, and bio-functional activity” describes the encapsulation of carvacrol into microcapsules via complex coacervation between walnut meal protein isolate (WMPI) and gum Arabic (GA) and their characterization in terms of particle size, loading capacity and encapsulation efficiency, Fourier transform infrared spectroscopy and fluorescent microscopy. The authors also describe the microcapsule release as well as the antioxidant and antibacterial activity.

The manuscript is overall well written, the data are appropriate, and the references are up to date. However, the authors should better clarify the originality of this work highlighting this aspect in the abstract as well as in the introduction section.

The authors should improve the discussion of the obtained results.

 Other comments

-Section 2.7.7.: the authors should introduce additional details regarding food simulant.

-Section 3.1.1.: the authors should check the accuracy of the sentence in Lines 227-228 “the z-potential of WMPI ranged from 21.53 mV (pH 3.0) to 10.43 mV (pH 6.0) and they should make sure that the results reported in the text fit with those shown in Figure 2 where the pH ranges from 3.0 to 5.5.

-Section 3.2.: the authors should discuss and compare the results obtained from the laser scattering size analyzer with those obtained from fluorescent microscopy. Moreover, Figure 4 should be improved.

-Section 3.3.: the authors should better discuss the results regarding the EE of Carvacrol also considering the equation 1 employed for the quantification.

-Figure 5A should be modified by better organizing the axes.

-Section 3.6.: the authors should improve the release discussion by taking into consideration the burst release.

-Section 3.8.: the authors could improve the quality of the manuscript by introducing additional antimicrobial assays to better clarify the potential antimicrobial activity of these systems. Moreover, the bacteria strains should be specified. Additionally, lines 406-407 are not clear, please clarify.

-The authors should introduce stability studies regarding the microcapsules.

Reviewer 3 Report

In this paper authors report research regarding carvacrol microcapsules preparation by complex coacervation between walnut meal protein isolate (WMPI) and gum Arabic (GA). Physicochemical properties of carvacrol microcapsules, antibacterial and antioxidant activities have been investigated and also the sustained release performance has been evaluated. The study is conducted using common methodology in the field. The manuscript is carefully conceived, interesting, clearly divided into sections and the results are well explained. However, there are some points that the authors should address:

-       -  The abbreviations used have to be fully given when they first appear (e.g. FAO in line 58)

-        -  Line 31 – “Carvacrol (5-Isopropyl-2-methylphenol), a natural essential oil obtained from oregano and thyme plants..”. Carvacrol is a major component of oregano and thyme oils, please reformulate.

-      -  The authors should discuss the effect of pH on turbidity

-    -  What was the basis for choosing the 2:1 carvacrol: biopolymers ratio? The authors should investigate the effect of wall:core ratio on encapsulation efficiency

-      -  References: please bold the years  

Round 2

Reviewer 2 Report

Authors improved the work considering the comments received but the manuscript, in my opinion, still needs minor changes before being accepted for the publication.

The authors should clarify the novelty of the study, also making comparisons between this work and literature data.

Section 2.7.7. This section is still unclear.

Section 3.3. authors introduced generical aspects regarding the EE. They also should introduce a discussion regarding the obtained results. In the figure 5A the y axis title is missing.

Section 3.6: the release discussion could be further improved by deeply describing the mechanism of release proposed by the authors.

Section and 3.8: authors generically refer to “another antimicrobial assay”, they should be more precise and support in the discussion the obtained results and how the formulation can act as antimicrobial. Authors should justify the bacterial strains employed in the study indicating the reason for their consideration.

Author Response

  1. The authors should clarify the novelty of the study, also making comparisons between this work and literature data.

Response: Suggestion was followed. We have added the description of the novelty of the study and made comparisons between this work and literature in the revised manuscript. The detailed revision can be found in Line 69-77, Page 2 in revised manuscript.

  1. Section 2.7.7. This section is still unclear.

Response: Suggestion was followed. We have further explained the release experiment regarding food simulant in the revised manuscript. The detailed revision can be found in Line 206-216, Page 5-6 in revised manuscript.

  1. Section 3.3. authors introduced generical aspects regarding the EE. They also should introduce a discussion regarding the obtained results. In the figure 5A the y axis title is missing.

Response: Suggestion was followed. We have added a discussion regarding the obtained results it in the revised manuscript. The detailed revision can be found in Line 337-341, Page 9 in revised manuscript. In addition, we have modified the y axis in the revised figure 5A. The y-axis corresponds to LC (%) and EE (%), respectively.

  1. Section 3.6: the release discussion could be further improved by deeply describing the mechanism of release proposed by the authors.

Response: Suggestion was followed. We have added the release discussion in the revised manuscript. The detailed revision can be found in Line 418-423, Page 12 in revised manuscript.

  1. Section and 3.8: authors generically refer to “another antimicrobial assay”, they should be more precise and support in the discussion the obtained results and how the formulation can act as antimicrobial. Authors should justify the bacterial strains employed in the study indicating the reason for their consideration.

Response: Suggestion was followed. We have added the discussion about the obtained results in another antimicrobial assay. The detailed revision can be found in Line 471-479, Page 13 in revised manuscript. In addition, we have addition the reason for the tested bacterial strains employed in this study. The detailed revision can be found in Line 451-454, Page 13 in revised manuscript. Finally, we further added other discussions on antibacterial capacity of the carvacrol microcapsules.

Reviewer 3 Report

In this revised form I would like to recommend the acceptance of the manuscript.

Author Response

Thanks for your suggestions and your time reviewing our manuscript.